# SDE Matching: Scalable and Simulation-Free Training of Latent Stochastic Differential Equations

**Grigory Bartosh** [1]  **Dmitry Vetrov** [2]  **Christian A. Naesseth** [1]

## Abstract

The Latent Stochastic Differential Equation (SDE) is a powerful tool for time series and sequence modeling. However, training Latent SDEs typically relies on adjoint sensitivity methods, which depend on simulation and backpropagation through approximate SDE solutions, which limit scalability. In this work, we propose SDE Matching, a new simulation-free method for training Latent SDEs. Inspired by modern Score- and Flow Matching algorithms for learning generative dynamics, we extend these ideas to the domain of stochastic dynamics for time series and sequence modeling, eliminating the need for costly numerical simulations. Our results demonstrate that SDE Matching achieves performance comparable to adjoint sensitivity methods while drastically reducing computational complexity.

## 1. Introduction

Differential equations are a natural choice for modeling continuous-time dynamical systems and have recently received significant interest in machine learning. Since Chen et al. (2018) introduced the adjoint sensitivity method for learning Ordinary Differential Equations (ODEs) in a memory-efficient manner, ODE-based approaches became popular in deep learning for density estimation (Grathwohl et al., 2019) and to model irregularly observed time series (Yildiz et al., 2019; Rubanova et al., 2019).

However, ODEs describe deterministic systems and encode all uncertainty into their initial conditions. This limits the applicability of ODE-based approaches when modeling stochastic and chaotic processes. To address these limitations, several works study Neural (or Latent) Stochastic

Table 1. Asymptotic complexity comparison. $L$ and $R$ are number of sequential and parallel evaluations of drift/diffusion terms, respectively, and $D$ is the number of parameters/states.

| Method | Memory | Time |
|---|---|---|
| Forward Pathwise (Yang & Kushner, 1991) (Gobet & Munos, 2005) | $\mathcal{O}(1)$ | $\mathcal{O}(LD)$ |
| Backprop through Solver (Giles & Glasserman, 2006) | $\mathcal{O}(L)$ | $\mathcal{O}(L)$ |
| Stochastic Adjoint (Li et al., 2020) | $\mathcal{O}(1)$ | $\mathcal{O}(L \log L)$ |
| Amortized Reparameterization (Course & Nair, 2023) | $\mathcal{O}(R)$ | $\mathcal{O}(R)$ |
| **SDE Matching** (this paper) | $\mathcal{O}(1)$ | $\mathcal{O}(1)$ |

Differential Equations (SDEs) (Movellan et al., 2002; Ha et al., 2018; Tzen & Raginsky, 2019; Peluchetti & Favaro, 2020; Hodgkinson et al., 2020). Latent SDEs have been shown to be more robust to data shifts (Oh et al., 2024), and applications are numerous, from neuroscience (ElGazzar & van Gerven, 2024) to video prediction (Daems et al., 2024).

To enable more memory-efficient learning, Li et al. (2020) extended the adjoint sensitivity method from ODEs to SDEs. Despite these advancements, training of ODE and SDE models is *simulation-based*, it relies on costly numerical integration of differential equations and backpropagation through the solutions. With training algorithms that are difficult to parallelize on modern hardware, Latent SDEs have till now resisted truly scaling.

In parallel, Score Matching (Ho et al., 2020; Song et al., 2021b) and Flow Matching methods (Lipman et al., 2023; Albergo et al., 2023; Liu et al., 2023) have demonstrated that continuous-time dynamics for generative modeling can be learned efficiently in a *simulation-free* manner—without requiring numerical integration. These techniques have proven computationally efficient and scalable for high-dimensional problems. Inspired by these developments, we develop a simulation-free approach for learning SDEs.

[1]University of Amsterdam [2]Constructor University, Bremen. Correspondence to: Grigory Bartosh <g.bartosh@uva.nl>, Dmitry Vetrov <dvetrov@constructor.university>, Christian A. Naesseth <c.a.naesseth@uva.nl>.

*Proceedings of the 42$^{nd}$ International Conference on Machine Learning*, Vancouver, Canada. PMLR 267, 2025. Copyright 2025 by the author(s).

We introduce SDE Matching—a simulation-free framework for learning Latent SDE models. The key idea we adopt from matching-based approaches is direct access to samples from the latent *posterior* approximation at any time step. This eliminates the need to integrate SDEs during training. The SDE Matching objective is estimated using the Monte Carlo method, achieving $\mathcal{O}(1)$ memory and time complexity. We summarize the complexity comparisons in Table 1.

We summarize our contributions as follows:

1. We establish a clear connection between diffusion models and Latent SDEs, which motivates the development of a more efficient, simulation-free training procedure.

2. We propose an efficient parameterization of the Latent SDE posterior process and use it to develop SDE Matching, a simulation-free training procedure for Latent and Neural SDEs.

3. We demonstrate that SDE Matching achieves comparable performance to the adjoint sensitivity method on both synthetic and real data, while significantly reducing the computational cost of training and improving convergence.

4. We show that SDE Matching enables the application of Latent SDEs to high-dimensional problems, where training with the adjoint sensitivity method is computationally infeasible.

In Section 2, we provide background on the Latent SDE model. In Section 3, we discuss the connection between Latent SDEs and matching-based methods, which serve as motivation for SDE Matching. Then, in Section 4, we introduce the parameterization of the Latent SDE posterior process and formulate the SDE Matching training algorithm. In Section 5, we demonstrate that our method achieves comparable performance on both synthetic and real-world data while requiring significantly less computational and memory cost for training. In Section 6 we discuss and contrast SDE Matching to related work. Finally, we conclude with a discussion of the limitations of our approach and potential directions for future work.

## 2. Background

Consider a series of observations $\mathbf{X} = x_{t_1}, x_{t_2}, \ldots, x_{t_N}$, where each $x_{t_i} \in \mathrm{X}$ where the space $\mathrm{X}$ depends on the application. For simplicity of notation, we assume that all time steps $t_i$ belong to the interval $[0, 1]$. The extension to intervals of arbitrary lengths is straightforward.

The Latent SDE model assumes that this series is generated constructively by the following stochastic process, referred to as the prior process. First, sample a latent variable

$z_0 \in \mathbb{R}^D$ from the initial prior distribution $p_\theta(z_0)$. Next, infer the latent continuous dynamics $\mathbf{Z} = (z(t))_{t \in [0,1]}$ by integrating the following SDE:

$$dz_t = h_\theta(z_t, t)dt + g_\theta(z_t, t)d\mathbf{w}, \tag{1}$$

where $h_\theta : \mathbb{R}^D \times [0, 1] \mapsto \mathbb{R}^D$ is the drift term, $g_\theta(z_t, t) : \mathbb{R}^D \times [0, 1] \mapsto \mathbb{R}^{D \times D}$ is the diffusion term, and $\mathbf{w}$ is a standard Wiener process. The SDE in Equation (1), together with the prior distribution $p_\theta(z_0)$, defines a sequence of probability distributions $p_\theta(z_t)$ at each time step $t \in [0, 1]$. Once the trajectory of the latent variables $\mathbf{Z}$ is sampled, we independently sample observations $x_{t_i}$ from the conditional distributions $p_\theta(x_{t_i}|z_{t_i})$ for each time step $t_i$.

The goal of the Latent SDE is to find a set of parameters $\theta$ that best fits a dataset of observed time series or sequences. Unfortunately, the posterior distribution of the latent variables, $p_\theta(\mathbf{Z}|\mathbf{X})$, is generally intractable. However, variational inference can be used to train the Latent SDE. Similar to the prior process, we introduce an approximate posterior process, which is conditioned on the observations $\mathbf{X}$. The posterior process consists of two components: the initial posterior distribution $q_\varphi(z_0|\mathbf{X})$ and a conditional SDE:

$$dz_t = f_\varphi(z_t, t, \mathbf{X})dt + g_\theta(z_t, t)d\mathbf{w}, \tag{2}$$

where $f_\varphi(z_t, t, \mathbf{X})$ defines the drift term of the SDE conditionally on the observations $\mathbf{X}$. It is important to note that, despite having a different drift term, the posterior shares the same diffusion term $g_\theta(z_t, t)$ as in Equation (1).

The training objective of the Latent SDE is a variational bound on the log-marginal likelihood of the observations:

$$-\log p_\theta(\mathbf{X}) \le \mathcal{L} = \mathcal{L}_{\mathrm{prior}} + \mathcal{L}_{\mathrm{diff}} + \mathcal{L}_{\mathrm{rec}} \tag{3}$$

$$\mathcal{L}_{\mathrm{prior}} = \mathrm{D}_{\mathrm{KL}}\big(q_\varphi(z_0|\mathbf{X})\|p_\theta(z_0)\big) \tag{4}$$

$$\mathcal{L}_{\mathrm{diff}} = \mathop{\mathbb{E}}_{q_\varphi(\mathbf{Z}|\mathbf{X})}\left[\int_0^1 \frac{1}{2}\|r_{\theta,\varphi}(z_t, t, \mathbf{X})\|_2^2\, dt\right] \tag{5}$$

$$\mathcal{L}_{\mathrm{rec}} = \mathop{\mathbb{E}}_{q_\varphi(\mathbf{Z}|\mathbf{X})}\left[\sum_{i=1}^N -\log p_\theta(x_{t_i}|z_{t_i})\right], \tag{6}$$

where $r_{\theta,\varphi}(z_t, t, \mathbf{X})$ satisfies

$$g_\theta(z_t, t)r_{\theta,\varphi}(z_t, t, \mathbf{X}) = h_\theta(z_t, t) - f_\varphi(z_t, t, \mathbf{X}). \tag{7}$$

Obtaining an estimate of the objective $\mathcal{L}$ for stochastic optimization requires joint numerical integration of the posterior process SDE (Equation (2)) and the integral in Equation (5). While there exist methods to improve the memory and computational efficiency of numerical integration, most are simulation-based and require backpropagation through numerical solutions of SDEs. Besides being computationally expensive, these methods are difficult to parallelize and suffer from numerical instabilities. Altogether, these challenges make training the Latent SDE a difficult task.

## 3. Diffusion Models as Latent SDEs

In contrast to Latent SDEs, recent advancements in diffusion and flow-based modeling demonstrate that continuous dynamics can be learned efficiently in a simulation-free manner, without requiring numerical integration. At first glance, these generative models seem quite different from Latent SDEs. Instead of generating data autoregressively, they produce a single data point through an iterative refinement process, gradually reconstructing corrupted data. However, similar to Latent SDEs, continuous diffusion models can be thought of as learning an SDE in a latent space. This connection is useful for understanding and developing SDE Matching.

Conventional diffusion models are defined through two processes: the forward (or noising) process and the reverse (or generative) process. The forward process takes a data point $x \in \mathbb{R}^D$ and perturbs it over time by injecting noise. This dynamic can be expressed as an SDE with a linear drift term and state-independent diffusion term:

$$dz_t = f(t)z_t dt + g(t)d\mathbf{w}, \qquad (8)$$

where $f : [0, 1] \mapsto \mathbb{R}$, $g : [0, 1] \mapsto \mathbb{R}_+$, and $q(z_0|x) \approx \delta(z_0 - x)$. Due to the linearity of Equation (8), the conditional marginal distribution is available in closed form $q(z_t|x) = \mathcal{N}(z_t; \alpha_t x, \sigma_t^2 I)$, where $\alpha_t, \sigma_t$ are determined by $f(t), g(t)$. The functions $f(t)$ and $g(t)$ are typically chosen to ensure that $q(z_1|x) \approx \mathcal{N}(z_1; 0, I)$. The generative process then reverses this transformation, starting from the prior $p(z_1) = \mathcal{N}(z_1; 0, I)$ and following the (marginal) reverse SDE:

$$dz_t = [f(t)z_t - g^2(t)\nabla_{z_t} \log q(z_t)]dt + g(t)d\bar{\mathbf{w}}. \quad (9)$$

Here, $\bar{\mathbf{w}}$ denotes a standard Wiener process, where time flows backward. Diffusion models approximate this reverse process by learning the score function $\nabla_{z_t} \log q(z_t)$ using a denoising score matching loss:

$$\mathop{\mathbb{E}}_{u(t)q(z_t|x)} \left[ \left\| s_\theta(z_t, t) - \nabla_{z_t} \log q(z_t|x) \right\|_2^2 \right], \qquad (10)$$

where $u(t)$ represents a uniform distribution over the interval $[0, 1]$, and $s_\theta : \mathbb{R}^d \times [0, 1] \mapsto \mathbb{R}^d$ is a learnable approximation of the score function.

A key property of the denoising score matching objective is that it can be learned in a simulation-free manner. Due to the simplicity of the forward process, instead of integrating the SDE in Equation (8), we can directly sample from $q(z_t|x)$ and estimate the expectation in Equation (10) using the Monte Carlo method. This property distinguishes diffusion models from Latent SDEs. However, as we will see diffusion models are in fact a special case of Latent SDEs.

To demonstrate this connection we: (1) derive the reverse SDE for the conditional forward process (Equation (8)), and

(2) invert the time direction for the entire model, mapping the time interval $[0, 1]$ into $[1, 0]$.

First, the reverse SDE for the forward process can be obtained from the Fokker–Planck equation, similar to the derivation of the marginal reverse SDE in Equation (9).

Second, after inverting the time direction, we obtain:

$$dz_t = [f(t)z_t + g^2(t)\nabla_{z_t} \log q(z_t|x)]dt + g(t)d\mathbf{w}. \quad (11)$$

Except for the change in time direction, the generative process remains unchanged:

$$dz_t = [f(t)z_t + g^2(t)s_\theta(z_t, t)]dt + g(t)d\mathbf{w}. \qquad (12)$$

From this perspective, diffusion models can be seen as a special case of Latent SDEs with a specific form of the posterior process and only a single observation $x$ at time step $t = 1$. Moreover, substituting this parameterization of the prior and reverse processes into the Latent SDE objective in Equation (3), we find that it corresponds to a reweighted denoising score matching objective:

$$\mathop{\mathbb{E}}_{q(\mathbf{Z}|x)} \left[ \int_0^1 \frac{1}{2g^2(t)} \left\| \cancel{f(t)z_t} + g^2(t)s_\theta(z_t, t) \right.\right.$$
$$\left.\left. - \cancel{f(t)z_t} - g^2(t)\nabla_{z_t} \log q(z_t|x) \right\|_2^2 dt \right] + C \quad (13)$$
$$= \mathop{\mathbb{E}}_{u(t)q(z_t|x)} \left[ \frac{g^2(t)}{2} \left\| s_\theta(z_t, t) - \nabla_{z_t} \log q(z_t|x) \right\|_2^2 \right] + C. \quad (14)$$

This result is particularly meaningful since it is well known that denoising score matching (Equation (10)) corresponds to a reweighted variational bound on the likelihood of diffusion models (Song et al., 2021a).

This connection between diffusion models and Latent SDEs, along with the fact that diffusion models can be trained in a simulation-free manner, motivates us to extend and develop a simulation-free framework for training Latent SDEs.

## 4. SDE Matching

Diffusion models can be trained in a simulation-free manner because they do not require full simulation of the noising process. Instead, they allow direct sampling of latent variables $z_t$ from the marginal distribution $q(z_t|x)$. In contrast, Latent SDEs are often trained using a posterior process parameterized by a general-form SDE. This renders the marginal posterior distribution $q_\varphi(z_t|\mathbf{X})$ intractable, which prevents simulation-free training.

To address this limitation, we propose SDE Matching – a framework for simulation-free training of Latent SDE models. The key idea behind SDE Matching is to parameterize the posterior process' conditional SDE via a learnable

function $F_\varphi(\varepsilon, t, \mathbf{X})$ that directly defines the marginal distributions $q_\varphi(z_t|\mathbf{X})$. This design inherently allows direct sampling of latent variables without numerical integration. Importantly, SDE Matching simplifies only the training procedure while leaving the generative dynamics of the Latent SDE prior process fully flexible.

### 4.1. Generative Model

In SDE Matching, the prior process is identical to the standard Latent SDE model (see Section 2):

$$dz_t = h_\theta(z_t, t)dt + g_\theta(z_t, t)d\mathbf{w}. \qquad (15)$$

In general, the functions $h_\theta$ and $g_\theta$ may be parameterized using neural networks of arbitrary form. However, the choice of parameterization involves some trade-offs, which we will discuss later.

Observations $x_{t_i}$ are likewise assumed to be sampled conditionally independently from the likelihood distributions $x_{t_i} \sim p_\theta(x_{t_i}|z_{t_i})$ for each time step $t_i$.

### 4.2. Posterior Process

Instead of defining the posterior process through a conditional SDE (Equation (2)) and then deriving its analytical solution, we propose an alternative approach. We first define the conditional marginal distribution of latent variables $q_\varphi(z_t|\mathbf{X})$ for each $t$, and then derive the corresponding conditional SDE with the desired marginal distributions.

We construct the posterior process by following three steps:

1. Define the marginal distribution implicitly through a parameterized function of noise;

2. Construct a conditional ODE that satisfies the target marginals;

3. Transition from the ODE to a SDE while preserving the marginal distributions.

A detailed description of each step with proofs is provided in Appendix A.

**Posterior Marginal Distribution.** We define the conditional marginal distribution $q_\varphi(z_t|\mathbf{X})$ implicitly. First, we introduce a function that, given time $t$ and observations $\mathbf{X}$, transforms noise $\varepsilon$ into a latent variable $z_t$:

$$z_t = F_\varphi(\varepsilon, t, \mathbf{X}), \qquad (16)$$

where $\varepsilon \sim q(\varepsilon) = \mathcal{N}(\varepsilon; 0, I)$. This implicitly defines the conditional distribution of latent variables $q_\varphi(z_t|\mathbf{X})$. Although, in general, we do not have an explicit form for this distribution, the above definition inherently enables efficient

sampling. The specific parameterization of $F_\varphi$ and the dependence of $z_t$ on observations $\mathbf{X}$ are user design choices.

**Conditional ODE.** We assume that $F_\varphi$ is differentiable with respect to $\varepsilon$ and $t$, and invertible with respect to $\varepsilon$. Fixing observations $\mathbf{X}$ and noise $\varepsilon$, and varying $t$ from 0 to 1, results in a smooth trajectory from $z_0$ to $z_1$. Differentiating these trajectories over time yields a velocity field that generates the conditional distribution $q_\varphi(z_t|\mathbf{X})$, the conditional ODE:

$$dz_t = \bar{f}_\varphi(z_t, t, \mathbf{X})dt, \quad \text{where} \qquad (17)$$

$$\bar{f}_\varphi(z_t, t, \mathbf{X}) = \left.\frac{\partial F_\varphi(\varepsilon, t, \mathbf{X})}{\partial t}\right|_{\varepsilon = F_\varphi^{-1}(z_t, t, \mathbf{X})}. \qquad (18)$$

Thus, if we sample $z_0 \sim q_\varphi(z_0|\mathbf{X})$ and solve Equation (17) up to time $t$, we obtain a sample $z_t \sim q_\varphi(z_t|\mathbf{X})$.

The time derivative of $F_\varphi$ can be computed efficiently using forward-mode differentiation in modern frameworks such as PyTorch (Paszke et al., 2017) or JAX (Bradbury et al., 2018).

**Conditional SDE.** Finally, we derive the conditional SDE that defines the posterior process.

Given access to both the conditional ODE and the score function $\nabla_{z_t} \log q_\varphi(z_t|\mathbf{X})$, we follow Song et al. (2021b) and define a conditional SDE with marginal distributions $q_\varphi(z_t|\mathbf{X})$ as follows:

$$dz_t = f_{\theta,\varphi}(z_t, t, \mathbf{X})dt + g_\theta(z_t, t)d\mathbf{w}, \quad \text{where} \qquad (19)$$

$$f_{\theta,\varphi}(z_t, t, \mathbf{X}) = \bar{f}_\varphi(z_t, t, \mathbf{X}) \qquad (20)$$

$$+ \frac{1}{2}g_\theta(z_t, t)g_\theta^\top(z_t, t)\nabla_{z_t} \log q_\varphi(z_t|\mathbf{X})$$

$$+ \frac{1}{2}\nabla_{z_t} \cdot \left[g_\theta(z_t, t)g_\theta^\top(z_t, t)\right].$$

Here, the diffusion term $g_\theta$ is the matrix-valued function from the prior process. It is crucial that the posterior process has the same diffusion term as the prior process to ensure that the variational bound is finite. Notably, $g_\theta$ affects only the distribution of trajectories $z(t)_{t \in [0,1]}$, while the marginal distributions $q_\varphi(z_t|\mathbf{X})$ do not depend on $g_\theta$.

We would like to emphasis that the correspondence between the posterior SDE in Equation (19) and its marginals $q_\varphi(z_t|\mathbf{X})$ is exact, not approximate.

Evaluating the drift term in Equation (19) presents several challenges. First, it requires access to the conditional score function $\nabla_{z_t} \log q_\varphi(z_t|\mathbf{X})$, which can be computationally expensive. However, for $F_\varphi$ functions that enable efficient log-determinant computation of the Jacobian matrix, the score function can be computed efficiently (e.g., functions linear in $\varepsilon$ or RealNVP architectures (Dinh et al., 2017; Kingma & Dhariwal, 2018)). The calculation of this score function is further discussed in Appendix A.4.

**Algorithm 1** Training of SDE Matching

**Require:** $\mathbf{X}, p_\theta(z_0), h_\theta, g_\theta, p_\theta(x_t|z_t), F_\varphi$
  **for** learning iterations **do**
    # Calculation of prior loss $\mathcal{L}_{\mathrm{prior}}$
    $\varepsilon_0 \sim \mathcal{N}(0, I)$
    $z_0 = F_\varphi(\varepsilon_0, 0, \mathbf{X})$
    $\mathcal{L}_{\mathrm{prior}} = \mathrm{D}_{\mathrm{KL}}\big(q_\varphi(z_0|\mathbf{X})\|p_\theta(z_0)\big)$
    # Calculation of diffusion loss $\mathcal{L}_{\mathrm{diff}}$
    $t \sim u(t)$
    $\varepsilon_t \sim \mathcal{N}(0, I)$
    $z_t = F_\varphi(\varepsilon_t, t, \mathbf{X})$
    $r_{\theta,\varphi}(z_t, t, \mathbf{X}) = g_\theta^{-1}(z_t, t)\left(h_\theta(z_t, t) - f_{\theta,\varphi}(z_t, t, \mathbf{X})\right)$
    $\mathcal{L}_{\mathrm{diff}} = \frac{1}{2} \|r_{\theta,\varphi}(z_t, t, \mathbf{X})\|_2^2$
    # Calculation of reconstruction loss $\mathcal{L}_{\mathrm{rec}}$
    $i \sim u(i)$
    $\varepsilon_{t_i} \sim \mathcal{N}(0, I)$
    $z_{t_i} = F_\varphi(\varepsilon_{t_i}, t_i, \mathbf{X})$
    $\mathcal{L}_{\mathrm{rec}} = -N \log p_\theta(x_{t_i}|z_{t_i})$
    # Optimization
    $\mathcal{L} = \mathcal{L}_{\mathrm{rec}} + \mathcal{L}_{\mathrm{diff}} + \mathcal{L}_{\mathrm{prior}}$
    Gradient step on $\theta$ and $\varphi$ w.r.t. $\mathcal{L}$
  **end for**

The second challenge involves computing the last term in Equation (20), which can be expensive in high-dimensional settings. However, it can be computed efficiently if for example the diffusion term $g_\theta$ is of the form $\sigma_\theta(z_t, t)\Gamma_\theta(t)$ for scalar-valued $\sigma_\theta$ and matrix-valued $\Gamma_\theta$. Another option is if $g_\theta$ is a diagonal matrix, where each element $g_\theta(z_t, t)_{k,k}$ depends only on the $k$-th coordinate of $z_t$. Furthermore, if $g_\theta$ does not depend on the state $z_t$, this term vanishes. Notably, while this structure limits flexibility, it is identical to constraints in memory-efficient implementations of standard Latent SDE training (Li et al., 2020), which means that SDE Matching does not introduce additional restrictions.

### 4.3. Optimization

Like standard Latent SDE training, SDE Matching optimizes the parameters $\theta$ and $\varphi$ end-to-end by optimizing the same variational objective. However, the training procedure of SDE Matching can be made simulation-free by leveraging the above posterior process and rewriting the objective from Equation (3) as follows:

$$-\log p_\theta(\mathbf{X}) \le \mathcal{L} = \mathcal{L}_{\mathrm{prior}} + \mathcal{L}_{\mathrm{diff}} + \mathcal{L}_{\mathrm{rec}} \quad (21)$$

$$\mathcal{L}_{\mathrm{prior}} = \mathrm{D}_{\mathrm{KL}}\big(q_\varphi(z_0|\mathbf{X})\|p_\theta(z_0)\big) \quad (22)$$

$$\mathcal{L}_{\mathrm{diff}} = \mathop{\mathbb{E}}_{u(t)q_\varphi(z_t|\mathbf{X})} \left[\frac{1}{2} \|r_{\theta,\varphi}(z_t, t, \mathbf{X})\|_2^2\right] \quad (23)$$

$$\mathcal{L}_{\mathrm{rec}} = N \mathop{\mathbb{E}}_{u(i)q_\varphi(z_{t_i}|\mathbf{X})} \left[-\log p_\theta(x_{t_i}|z_{t_i})\right], \quad (24)$$

where $u(t)$ represents a uniform distribution over the inter-

val $[0, 1]$, $u(i)$ represents a uniform distribution over the set $1, \ldots, N$, and $r_{\theta,\varphi}(z_t, t, \mathbf{X})$ satisfies:

$$g_\theta(z_t, t)r_{\theta,\varphi}(z_t, t, \mathbf{X}) = h_\theta(z_t, t) - f_{\theta,\varphi}(z_t, t, \mathbf{X}). \quad (25)$$

The objective in Equation (21) consists of three terms: the prior loss $\mathcal{L}_{\mathrm{prior}}$, the diffusion loss $\mathcal{L}_{\mathrm{diff}}$, and the reconstruction loss $\mathcal{L}_{\mathrm{rec}}$. The optimization of $\mathcal{L}_{\mathrm{prior}}$ with respect to parameters $\theta$ can be omitted if we are not interested in unconditional sampling or in learning the initial prior $p_\theta(z_0)$—for example, if we are solely interested in forecasting. Otherwise, this term can still be optimized efficiently.

The other two terms, $\mathcal{L}_{\mathrm{diff}}$ and $\mathcal{L}_{\mathrm{rec}}$, which previously required the numerical integration of the conditional SDE, can now be estimated more efficiently. Specifically, for each term, we can first sample a timestep $t$, then sample $z_t \sim q_\varphi(z_t|\mathbf{X})$, and finally evaluate the corresponding function. Importantly, SDE Matching allows the estimation of $\mathcal{L}_{\mathrm{diff}}$ and $\mathcal{L}_{\mathrm{rec}}$ with an arbitrary number of samples evaluated in parallel. It also enables inference of the posterior process (Equation (19)) as a special case. However, we found that using a single sample was sufficient for our experiments. We summarize the training procedure in Algorithm 1.

Note that while we use the functions $\bar{f}_\varphi$ (Equation (17)) and $f_{\theta,\varphi}$ (Equation (19)) to describe the conditional SDE in the posterior process, these functions are ultimately defined by the reparameterization function $F_\varphi$ (Equation (16)) and $g_\theta$, which are the functions we actually parameterize.

The full Bayesian treatment is also possible as a straightforward extension of the bound in Equation (21) for a given prior with a separate variational approximation for $\theta$.

We would like to emphasize that the SDE Matching objective is exactly the same variational bound as used in the adjoint sensitivity method. The key difference lies in the estimation strategy for the objective. We discuss the tightness of the variational bound in Equation (21) in Appendix B and empirically validate convergence to the true posterior for a linear SDE in Appendix C.1.

### 4.4. Sampling

Unconditional sampling at inference- or test-time for SDE Matching follows the same procedure as sampling from a Latent SDE. First, we sample an initial latent variable $z_0 \sim p_\theta(z_0)$ and then integrate the unconditional SDE in Equation (15) using any off-the-shelf SDE solver. Then, at each desired time step $t$, we project the latent states $z_t$ to the data space by sampling $x_t$ from the distribution $p_\theta(x_t|z_t)$.

However, if we have access to partial observations $\mathbf{X} = x_{t_1}, x_{t_2}, \ldots, x_{t_N}$ and are interested in forecasting for $t > t_N$, we can instead first sample the latent state $z_{t_N}$ from $q_\varphi(z_{t_N}|\mathbf{X})$ and then integrate the prior process dynamics

using this sample as initial condition. This follows because the Latent SDE is Markovian with observations that are conditionally independent given the latent state. Importantly, because SDE Matching provides direct access to an approximation to the posterior (smoothing) marginals it can be used for simulation-free sampling (inference) of the latent state, whereas conventional parameterization of the posterior process would require simulating the conditional SDE up to time step $t_N$ first.

Similarly, interpolation can be performed by inferring only the posterior process dynamics. Alternatively, for more general conditioning events and steering one could leverage Sequential Monte Carlo methods (Naesseth et al., 2019; Chopin et al., 2020; Wu et al., 2023) for conditional sampling, but a detailed investigation of these approaches is beyond the scope of this work.

Due to the strong connection between SDE Matching and diffusion models—and given that diffusion models have demonstrated exceptional performance in deterministic sampling—it is natural to ask whether it is possible to sample deterministically from SDE Matching. Indeed, the prior process and the initial distribution $p_\theta(z_0)$ together define a sequence of marginal probability distributions $p_\theta(z_t)$. While it is possible to learn an ODE corresponding to $p_\theta(z_t)$, it is important to clarify that solving this ODE does not necessarily produce meaningful time-series samples.

Although the ODE preserves marginal distributions, it alters the joint distribution of latent variable trajectories. The same applies to the observed variables: while the marginal distributions remain correct, the joint distribution becomes distorted. Therefore, using an ODE for deterministic sampling of time series data does not generally preserve the correct structure of the generated trajectories.

## 5. Experiments

SDE Matching is a simulation-free framework for training Latent SDE models. Therefore, the goal of this section is to demonstrate that SDE Matching indeed enables computationally efficient training of Latent SDEs. Our objective is not to outperform all other Latent SDE-based approaches for time series modeling. Instead, SDE Matching is compatible with existing extensions and can be combined with them for further improvements.

To this end, we provide various experiments on both synthetic and real data. In all experiments, we use the same hyperparameters for both the conventional parameterization of Latent SDE and the parameterization described in Section 4. Additional details on parameterization and training is provided in Appendix D. When using the SDE Matching training procedure, we jointly optimize the parameters of the generative model $\theta$ in the prior $p_\theta(z_0)$, drift $h_\theta(z_t, t)$,

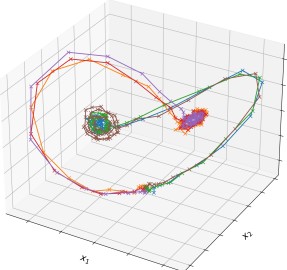

(a) Training data distribution

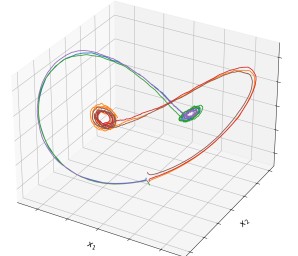 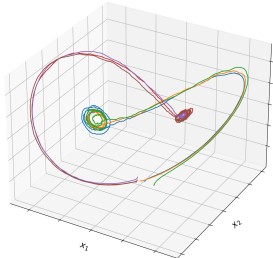

(b) Adjoint sensitivity method     (c) SDE Matching

*Figure 1.* Training data distribution and learned dynamics from a 3D stochastic Lorenz attractor. Results for Latent SDE trained with adjoint sensitivity method (*bottom left*) and SDE Matching (*bottom right*). Both methods successfully learn the underlying dynamics.

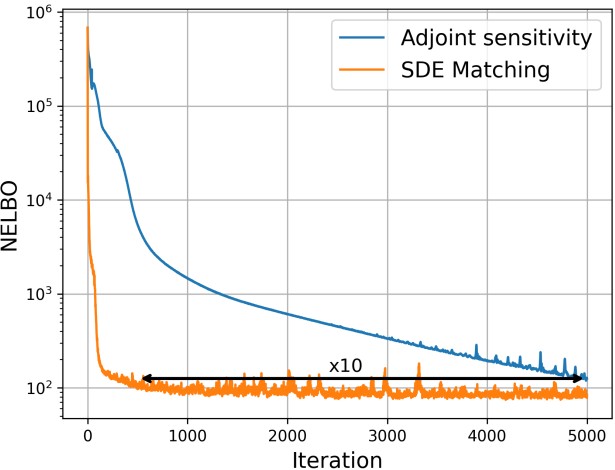

*Figure 2.* NELBO (Equation (21)) objective with respect to iteration step on 3D stochastic Lorenz attractor data. Results for Latent SDE trained with adjoint sensitivity method and SDE Matching. SDE Matching demonstrates approximatly 10 times faster convergence in terms of iterations.

diffusion $g_\theta(z_t, t)$, and observation model $p_\theta(x_t|z_t)$, as well as the parameters $\varphi$ of the posterior reparameterization function $F_\varphi$. All parameters are optimized jointly by minimizing

the variational bound in Equation (21). We do not apply annealing or additional regularization techniques during training.

The experiments demonstrate that SDE Matching achieves comparable or better accuracy than simulation-based Latent SDE training, while significantly reducing computational complexity. Not only does SDE Matching reduce the computational cost of each training iteration, the parameterization of the posterior process also leads to faster convergence in terms of the number of iterations for further gains compared to alternatives. The findings suggest that SDE Matching enhances the scalability of Latent SDEs to longer and higher-dimensional time series.

The code is available at https://github.com/GrigoryBartosh/sde_matching.

### 5.1. Synthetic Datasets

For the synthetic data experiment, we consider the 3D stochastic Lorenz attractor process from Li et al. (2020) with identical observation generation procedure.

As shown in Figure 1, both models—one trained using the adjoint sensitivity method and the other with SDE Matching—successfully learned the underlying dynamics. However, SDE Matching required only a single evaluation of the drift term in the posterior process for each iteration, whereas the adjoint sensitivity method required 100 simulation steps for this experiment. In terms of absolute runtime, a single training iteration with SDE Matching was approximately five times faster.

Moreover, as demonstrated in Figure 2, SDE Matching leads to faster convergence of the model. We attribute this to the parameterization of the posterior process through the reparameterization function $F_\varphi$ (Equation (16)). We hypothesize that since this function directly models the marginal distributions of the posterior process, the model can more efficiently learn compared to integrating conditional SDEs.

The combined, per-iteration and convergence, runtime speed-up is over $50\times$ compared to the adjoint sensitivity approach for this experiment.

To evaluate robustness with respect to time horison, we compare the gradient norms of the objective function with respect to model parameters for both SDE Matching and the adjoint sensitivity method. When integrating over time horizons $T \in 1, 2, 5, 10$, the adjoint sensitivity method yields the following $\log_{10}$-based gradient norms: $6.26 \pm 0.14$, $6.95 \pm 0.20$, $7.91 \pm 0.23$ and $8.82 \pm 0.28$. These results demonstrate that the gradient norms grow exponentially with the time horizon for adjoint sensitivity methods.

In contrast, SDE Matching maintains a stable gradient norm of $4.92 \pm 0.24$ (in $\log_{10}$ scale) across all time horizons,

*Table 2.* Test MSE on motion capture dataset. We report an average performance based on 10 models trained with deferent random seeds and 95% confidence interval based on t-statistic. The first section of the table contains simulation-based approaches and the second part contains simulation-free approaches. [†]results from (Yildiz et al., 2019), [*]from (Li et al., 2020) and [‡]from (Course & Nair, 2023).

| Model | Test MSE ↓ |
|---|---|
| DTSBN-S (Gan et al., 2015) | $34.86 \pm 0.02$[†] |
| npODE (Heinonen et al., 2018) | $22.96$[†] |
| NeuralODE (Chen et al., 2018) | $22.49 \pm 0.88$[†] |
| ODE$^2$VAE (Yildiz et al., 2019) | $10.06 \pm 1.4$[†] |
| ODE$^2$VAE-KL (Yildiz et al., 2019) | $8.09 \pm 1.95$[†] |
| Latent ODE (Rubanova et al., 2019) | $5.98 \pm 0.28$[*] |
| Latent SDE (Li et al., 2020) | $\mathbf{4.03 \pm 0.2}$[*] |
| ARCTA (Course & Nair, 2023) | $7.62 \pm 0.93$[‡] |
| SDE Matching (ours) | $\mathbf{4.50 \pm 0.32}$ |

indicating significantly greater stability for long time series modeling.

In Appendix C.2 we provide additional experiments demonstrating robustness of the SDE Matching training procedure.

### 5.2. Motion Capture Dataset

To validate SDE Matching on real-world data, we follow Li et al. (2020); Course & Nair (2023) and evaluate it on the motion capture dataset from Gan et al. (2015). This dataset consists of motion recordings from subject 35 walking, represented as 50-dimensional time series with 300 observations each. The dataset is split into 16 training sequences, 3 validation sequences, and 4 test sequences, following the preprocessing method from Wang et al. (2007).

We use the same hyperparameters as Li et al. (2020), including setting the latent state dimensionality to 6 and keeping the number of training iterations unchanged. In Table 2, we report the average performance on the training set over 10 models trained from different random seeds. SDE Matching achieves similar performance to the adjoint sensitivity method while being significantly more computationally efficient.

SDE Matching also outperforms the other simulation-free approach, ARCTA Course & Nair (2023), which can be seen as a special case of SDE Matching. Notably, ARCTA requires drawing around 100 latent samples and evaluations of SDEs per batch, whereas SDE Matching requires only one. We attribute this result to the fact that ARCTA employs a less flexible prior and posterior parameterization, where the volatility function $g_\theta$ is state-independent and distribution of latent states $q_\varphi(z_t|\mathbf{X})$ depends heavily on observations close to time $t$. This dependence makes it

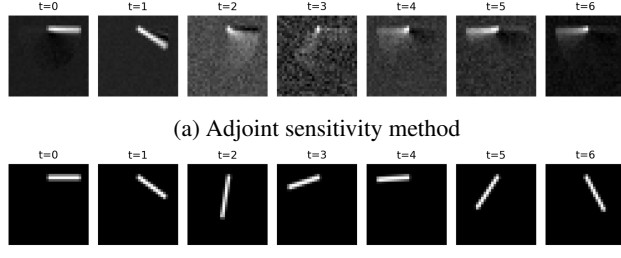

(a) Adjoint sensitivity method

(b) SDE Matching

*Figure 3.* Qualitative comparison of generated video sequences depicting a moving pendulum. Samples generated by the model trained with adjoint sensitivity method and SDE Matching with same number of training iterations.

challenging to learn long-term dependencies and limits the ability to propagate learning signals effectively.

An extended discussion on the importance of state-dependent volatility functions $g_\theta$, along with experiments on the state-dependent stochastic Lotka-Volterra system, is provided in Appendix C.3.

### 5.3. Video Dataset

Finally, we evaluate the performance of SDE Matching in a high-dimensional setting by modeling sequences of $32 \times 32$ images—i.e. video data—depicting a moving pendulum.

Using an identical number of training iterations (20k), SDE Matching successfully learns the underlying dynamics in approximately 20 minutes on a single GPU. In contrast, the adjoint sensitivity method requires around 30 hours for the same number of iterations and still fails to accurately capture the system's dynamics.

Visualizations of the generated sequences are provided in Figure 3.

## 6. Related Work

Differential equations are a widely recognized technique for modeling continuous dynamics. Recently they have seen a surge of interest in machine learning after the introduction of Neural ODEs by Chen et al. (2018) and Rubanova et al. (2019). Neural ODEs demonstrated strong performance in time-series modeling compared to recurrent neural networks, particularly for irregularly sampled time series. However, Neural ODEs have several limitations.

First, they rely on adjoint sensitivity methods for training, which requires numerical integration of gradients and back-propagation through the solutions. Aside from being computationally expensive and difficult to parallelize on modern hardware, ODEs can sometimes suffer from numerical instabilities (Lea et al., 2000). Second, ODE-based models inher-

ently encode all uncertainty into the initial conditions as the dynamics is completely deterministic. This approach is inadequate for modeling fundamentally stochastic processes, especially when observations are sparse or uninformative.

Traditionally, Monte Carlo methods have been used to train SDEs (Movellan et al., 2002; Beskos et al., 2006; van der Meulen & Schauer, 2017). Another line of work (Archambeau et al., 2007a;b) has focused on variational inference techniques for inferring the Latent SDEs's training objective. Discretization is used by Ryder et al. (2018) to enable more expressive variational approximations, and Ha et al. (2018) draw the connection to stochastic optimal control for more flexibility.

More recently, Li et al. (2020) extended the adjoint sensitivity method from ODEs to SDEs. Unlike Neural ODEs, Latent SDEs are better suited for modeling inherently stochastic and chaotic processes and are more robust to data shifts.

Many studies have proposed modifications to objective functions, introduced regularized dynamics, and improved computational efficiency and numerical stability for both Neural ODEs (Kelly et al., 2020; Finlay et al., 2020; Kidger et al., 2021a) and Latent SDEs (Kidger et al., 2021b). However, most of these methods still rely on backpropagation through the numerical solutions of differential equations limiting their scalability. In Verma et al. (2024) the authors instead rely on a site-based Gaussian approximation and suggest a fixed-point training algorithm.

Closest to our work is Course & Nair (2023) that develops a method for training Latent SDEs in a simulation-free manner. This approach is restricted to posterior processes with Gaussian marginals and does not support state-dependent diffusion terms in the prior process. Additionally, this method assumes that the latent state in the posterior process depends only on a few temporally closest observations. As a result, for effective optimization it requires drawing approximately 100 latent samples, rather than 1, per batch during training. This method can be seen as a special case of SDE Matching with the corresponding restrictions on the parameterization of the prior process and posterior process.

Recently, Zhang et al. (2024) proposed a simulation-free technique for time-series modeling. However, this approach learns dynamics directly in the original data space rather than in a latent space. Furthermore, to model non-Markovian processes the authors condition the generative dynamics on a fixed set of past observations, making simulations more computationally expensive and limiting the model's ability to capture long-range dependencies. This can be viewed as a special case of SDE Matching, where the latent space is constructed by concatenation of the most recent observations.

Another line of research that explores learning stochastic

dynamics in a simulation-free manner is diffusion models (Ho et al., 2020; Song et al., 2021b). As we demonstrated in Section 3, conventional diffusion models can be seen as a special case of Latent SDEs with only a single observation. The key property that enables efficient training in conventional diffusion models is their simple and fixed noising process, i.e., the posterior process.

Recently Singhal et al. (2023); Bartosh et al. (2023); Nielsen et al. (2024); Sahoo et al. (2024); Bartosh et al. (2024); Du et al. (2024) have shown that the noising process in diffusion models can be more complex, and even learnable, while still preserving the simulation-free property. These approaches can also be viewed as special cases of SDE Matching by the re-interpretation of diffusion models as Latent SDEs.

## 7. Limitations and Future Work

The simulation-free properties of SDE Matching come with certain trade-offs. First, SDE Matching parameterizes the posterior process through the function $F_\varphi(\varepsilon, t, \mathbf{X})$ (Equation (16)). This function must not only be smooth and invertible with respect to $\varepsilon$, it must also provide access to the conditional score function $\nabla_{z_t} \log q_\varphi(z_t | \mathbf{X})$ (Equation (20)). These requirements restrict the flexibility of $F_\varphi$ and, consequently, the posterior distribution approximation. Nevertheless, to the best of our knowledge, SDE Matching offers the most flexible parameterization that enables simulation-free access to latent samples $z_t \sim q_\varphi(z_t | \mathbf{X})$ of the conditional SDE posterior process.

Another limitation is the computational cost of evaluating the posterior process' SDE drift term for high-dimensional $g_\theta(z_t, t)$ (Equation (15)) of general form, as discussed in Section 4.2. However, it is worth reiterating that other memory-efficient training of Latent SDEs faces this same limitation.

SDE Matching does not constrain the flexibility of the prior process, which defines the generative model. Nevertheless, developing methods that allow even more flexible parameterization of $F_\varphi$ and diffusion terms $g_\theta$ remains a promising direction for future research. Additionally, understanding the impact of this flexibility in modeling latent dynamics versus dynamics in the original data space is an interesting open question.

We believe that thanks to its simulation-free properties, SDE Matching has the potential to scale Latent SDE-based modeling to high-dimensional applications like audio and video generation and many other applications where they were previously infeasible. It could potentially also contribute to the development of simulation-free methods for training Latent ODE models and learning policies in reinforcement learning. We leave these directions for future work.

## 8. Conclusion

We introduced SDE Matching, a simulation-free framework for training Latent SDEs. By leveraging insights from score-based generative models, we formulated a method that eliminates the need for costly numerical simulations while maintaining the expressiveness of Latent SDEs. Our approach directly parameterizes the marginal posterior distributions, enabling efficient training and conditional inference.

Through both synthetic and real-world experiments, we demonstrated that SDE Matching achieves performance comparable to or better than existing adjoint sensitivity-based methods, while significantly reducing computational complexity. The ability to estimate latent trajectories without solving high-dimensional SDEs makes our method particularly suitable for large-scale time-series and sequence modeling.

Despite its advantages, SDE Matching introduces trade-offs in terms of posterior parameterization flexibility. Nevertheless, to the best of our knowledge, SDE Matching proposes the most flexible parameterization of the posterior process that allows simulation-free sampling of latent variables. Exploring more expressive function classes for posterior distributions and further extending SDE Matching are promising directions for future work.

Furthermore, our approach provides a foundation for scaling Latent SDEs to previously infeasible sequential domains such as audio and video. We believe that the scalability and efficiency of SDE Matching will enable broader applications in scientific modeling, finance, and healthcare, where structured uncertainty modeling is critical. Future research may also explore extensions to simulation-free training of Latent ODEs and connections to reinforcement learning and stochastic optimal control.

## Impact Statement

This paper presents work whose goal is to advance the field of Machine Learning. There are many potential societal consequences of our work, none which we feel must be specifically highlighted here.

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

# A. Detailed Description of the Posterior Process

## A.1. Marginal Distribution

The posterior marginal distribution is defined by a transformation $F_\varphi$ from noise $\varepsilon$ for each $t$ and $\mathbf{X}$:

$$z_t = F_\varphi(\varepsilon, t, \mathbf{X}), \quad \varepsilon \sim q(\varepsilon) = \mathcal{N}(\varepsilon; 0, I). \tag{26}$$

Assuming that $F_\varphi$ is invertible and differentiable in $\varepsilon$ for each $t$ and $\mathbf{X}$. Then, we can use the standard change of variables result for distributions (Rudin, 2006; Bogachev, 2007) to derive the density of $z_t$

$$q_\varphi(z_t|\mathbf{X}) = q(\varepsilon)\Big|_{\varepsilon = F_\varphi^{-1}(z_t, t, \mathbf{X})} \cdot \left| \frac{\partial F_\varphi^{-1}(z_t, t, \mathbf{X})}{\partial z_t} \right|, \tag{27}$$

where $F_\varphi^{-1}$ is the inverse of $F_\varphi$ and $|\cdot|$ denotes the absolute value of the determinant. This is a well-known result of normalizing flows (Papamakarios et al., 2021) applied to our setting.

## A.2. Conditional ODE

We leverage results based on *flows*, solutions to ODEs, inspired by Biloš et al. (2021); Lee (2012); Bartosh et al. (2024) to turn a flow $F_\varphi$ into its corresponding generating ODE $\bar{f}_\varphi$.

**Proposition A.1.** *Let* $F_\varphi : \mathbb{R}^D \times \mathbb{R} \times \mathrm{X} \mapsto \mathbb{R}^D$ *be a smooth function such that* $F_\varphi(\cdot, t, \mathbf{X})$ *is invertible* $\forall t, \mathbf{X}$. *Then,* $F_\varphi(\cdot, t, \mathbf{X})$ *is a* flow *with the infinitesimal generator*

$$\frac{dF_\varphi(\varepsilon, t, \mathbf{X})}{dt}, \tag{28}$$

*which is a smooth vector field* $f$ *such that*

$$\frac{\partial F_\varphi(\varepsilon, t, \mathbf{X})}{\partial t} = f(F_\varphi(\varepsilon, t, \mathbf{X}), t, \mathbf{X}). \tag{29}$$

*Proof.* The result is a consequence of the Fundamental Theorem on Flows (Lee, 2012, Theorem 9.12) for the flow defined by $F_\varphi(\cdot, t, \mathbf{X})$. $\qquad\square$

Using Proposition A.1 with $\varepsilon = F_\varphi^{-1}(z_t, t, \mathbf{X})$, identifying that $F_\varphi(F_\varphi^{-1}(z_t, t, \mathbf{X}), t, \mathbf{X}) = z_t$, lets us construct the conditional ODE whose solution matches those of the flow $z_t = F_\varphi(\varepsilon, t, \mathbf{X})$:

$$dz_t = \bar{f}_\varphi(z_t, t, \mathbf{X})dt, \quad \text{where} \tag{30}$$

$$\bar{f}_\varphi(z_t, t, \mathbf{X}) = \frac{\partial F_\varphi(\varepsilon, t, \mathbf{X})}{\partial t}\bigg|_{\varepsilon = F_\varphi^{-1}(z_t, t, \mathbf{X})}. \tag{31}$$

Initializing using $z_0 \sim q_\phi(z_0|\mathbf{X})$ and solving the above conditional ODE in Equation (30) ensures $z_t \stackrel{d}{=} F_\varphi(\varepsilon, t, \mathbf{X})$ for $\varepsilon \sim q(\varepsilon)$, where $\stackrel{d}{=}$ denotes equal in distribution.

## A.3. Conditional SDE

To derive the conditional SDE we leverage a result by Song et al. (2021b) that relates the marginal distributions of an SDE with its corresponding (probability flow) ODE, restated in Proposition A.2 for convenience.

**Proposition A.2.** *The marginal distributions* $p_t(z_t)$ *of the SDE*

$$dz_t = h(z_t, t)dt + g(z_t, t)d\mathbf{w}, \tag{32}$$

*are, under suitable conditions on the drift and diffusion terms* $h$ *and* $g$*, identical to the distributions induced by the ODE*

$$dz_t = \left( h(z_t, t) - \frac{1}{2}g(z_t, t)g^\top(z_t, t)\nabla_{z_t} \log p_t(z_t) - \frac{1}{2}\nabla_{z_t} \cdot \left[ g(z_t, t)g^\top(z_t, t) \right] \right) dt. \tag{33}$$

*Proof.* See Song et al. (2021b, Appendix D.1). □

Note that this equivalence lets us easily find the corresponding (probability flow) ODE. However, we can equally well use it in the reverse order with a known ODE to construct the corresponding SDE by selecting a diffusion term $g$ and using Equation (33) to compute the drift term $h$.

For the prior process diffusion term $g_\theta(z_t, t)$ with velocity field Equation (30) we apply Proposition A.2 to find the conditional SDE with marginals $q_\varphi(z_t|\mathbf{X})$

$$dz_t = f_{\theta,\varphi}(z_t, t, \mathbf{X})dt + g_\theta(z_t, t)d\mathbf{w}, \quad \text{where} \tag{34}$$

$$f_{\theta,\varphi}(z_t, t, \mathbf{X}) = \bar{f}_\varphi(z_t, t, \mathbf{X}) + \frac{1}{2}g_\theta(z_t, t)g_\theta^\top(z_t, t)\nabla_{z_t}\log q_\varphi(z_t|\mathbf{X}) + \frac{1}{2}\nabla_{z_t} \cdot \left[g_\theta(z_t, t)g_\theta^\top(z_t, t)\right]. \tag{35}$$

This result follows by simply identifying terms in Proposition A.2 with our specific case

$$h(z_t, t) \equiv f_{\theta,\varphi}(z_t, t, \mathbf{X}),$$
$$g(z_t, t) \equiv g_\theta(z_t, t).$$

### A.4. Parameterization

As discussed in Section 4.2, evaluating the posterior SDE (Equation (19)) requires access to the conditional score function $\nabla_{z_t}\log q_\varphi(z_t|\mathbf{X})$. Since we parameterize the posterior marginals $q_\varphi(z_t|\mathbf{X})$ implicitly through an invertible transformation $F_\varphi$ (Equation (16)) of a random variable $\varepsilon \sim q(\varepsilon)$ into $z_t$, the score function can in general be computed using Equation (27):

$$\nabla_{z_t}\log q_\varphi(z_t|\mathbf{X}) = \nabla_{z_t}\left[\log q(\varepsilon)\Big|_{\varepsilon=F_\varphi^{-1}(z_t,t,\mathbf{X})} + \log\left|\frac{\partial F_\varphi^{-1}(z_t,t,\mathbf{X})}{\partial z_t}\right|\right]. \tag{36}$$

The first term in Equation (36) represents the log-density of the noise distribution, which is straightforward to compute. The second term is the log-determinant of the Jacobian matrix of the inverse transformation. If this Jacobian is available, the score function can be efficiently computed using automatic differentiation tools.

In this work, we parameterize the function $F_\varphi$ as follows:

$$F_\varphi(\varepsilon, t, \mathbf{X}) = \mu_\varphi(\mathbf{X}, t) + \sigma_\varphi(\mathbf{X}, t)\varepsilon, \quad \text{and} \quad F_\varphi^{-1}(z_t, t, \mathbf{X}) = \sigma_\varphi^{-1}(\mathbf{X}, t)\big(z_t - \mu_\varphi(\mathbf{X}, t)\big) \tag{37}$$

where $\sigma_\varphi$ is a matrix valued function.

Substituting this parameterization into Equation (36), the score function simplifies to:

$$\nabla_{z_t}\log q_\varphi(z_t|\mathbf{X}) = \left[\sigma_\varphi(\mathbf{X}, t)\sigma_\varphi^\top(\mathbf{X}, t)\right]^{-1}\big(\mu_\varphi(\mathbf{X}, t) - z_t\big) = -\sigma_\varphi^{-\top}(\mathbf{X}, t)\varepsilon\Big|_{\varepsilon=F_\varphi^{-1}(z_t,t,\mathbf{X})}. \tag{38}$$

The SDE Matching framework allows for other, potentially more flexible, parameterizations of the function $F_\varphi$. However, we leave the exploration of such alternatives for future research.

## B. Theoretical Considerations

In this section, we provide an extended discussion on the connection between the prior and posterior processes. Our primary goal is to address the question: under what conditions does the variational bound in Equation (21) become tight and match the true likelihood of the model?

The variational bound becomes tight when the approximate posterior distribution exactly matches the true posterior. In the case of the Latent SDE, this means that the posterior process must match the true posterior of the prior process.

The prior process (Equation (15)) is given by:

$$dz_t = h_\theta(z_t, t)dt + g_\theta(z_t, t)d\mathbf{w}. \tag{39}$$

Given a series of observations $\mathbf{X}$, the true posterior process can be derived via Doob's $h$-transform (Mider et al., 2021; Park et al., 2024), yielding:

$$dz_t = \left[ \underbrace{h_\theta(z_t, t) + g_\theta(z_t, t)g_\theta^\top(z_t, t)\nabla_{z_t} \log p_\theta(\mathbf{X}_t|z_t)}_{h_\theta(z_t, t, \mathbf{X})} \right] dt + g_\theta(z_t, t)d\mathbf{w}, \tag{40}$$

where $\mathbf{X}_t = \{x_s : x_s \in \mathbf{X}, s \geq t\}$ denotes the set of future observations from time $t$ onward.

We can also consider the corresponding deterministic process that follows the posterior marginals $p_\theta(z_t|\mathbf{X})$

$$\frac{dz_t}{dt} = \bar{h}_\theta(z_t, t, \mathbf{X}) \tag{41}$$

$$= h_\theta(z_t, t) + g_\theta(z_t, t)g_\theta^\top(z_t, t) \left[ \nabla_{z_t} \log p_\theta(\mathbf{X}_t|z_t) - \frac{1}{2}\nabla_{z_t} \log p_\theta(z_t|\mathbf{X}) \right] - \frac{1}{2}\nabla_{z_t} \cdot \left[ g_\theta(z_t, t)g_\theta^\top(z_t, t) \right] \tag{42}$$

Now, consider the approximate posterior process (Equation (19)) of the form:

$$dz_t = f_{\theta,\varphi}(z_t, t, \mathbf{X})dt + g_\theta(z_t, t)d\mathbf{w}. \tag{43}$$

The approximate process matches the true posterior process if the drift terms are equal: $f_{\theta,\varphi}(z_t, t, \mathbf{X}) \equiv h_\theta(z_t, t, \mathbf{X})$ or equivalently, if the corresponding deterministic processes (Equations (17) and (42)) match: $\bar{f}_{\theta,\varphi}(z_t, t, \mathbf{X}) \equiv \bar{h}_\theta(z_t, t, \mathbf{X})$. In terms of the reparameterization function $F_\varphi(\varepsilon, t, \mathbf{X})$ (Equation (16)) that defines the approximate posterior process, tthe variational bound becomes tight, and the approximate posterior process matches the true posterior, if $F_\varphi$ learns the solution trajectories of the ODE defined by $\bar{h}_\theta(z_t, t, \mathbf{X})$ (Equation (42)).

Unfortunately, as is typical in variational inference, the true posterior is intractable. Nevertheless, its analytical form provides valuable insight into the behavior that the approximate posterior should approximate. Therefore, with a sufficiently flexible reparameterization function $F_\varphi(\varepsilon, t, \mathbf{X})$, the SDE Matching approach should, in principle, be capable of learning the true posterior process and the variational bound becomes equal to the negative log-marginal likelihood.

## C. Additional Results

### C.1. Convergence to Likelihood

To empirically validate the tightness of the ELBO in SDE Matching, as discussed in Appendix B, we set up a linear stochastic system of the form:

$$dz_t = F(t)z_t dt + L(t)d\mathbf{w} \tag{44}$$
$$x_t = H(t)z_t + r_t, \tag{45}$$

where $r_t \sim \mathcal{N}(0, R(t))$. For this system, the true log-marginal likelihood can be computed using the Kalman filter (Sarkka & Solin, 2019).

In this experiment, we use simple coefficients: $F(t) = -t$, $L(t) = t$, $H(t) = 1$, and $R(t) = 0.01$. The model is parameterized identically to our experiment on the 3D stochastic Lorenz attractor (Section 5.1). As shown in Appendix C.3, the SDE Matching ELBO closely converges to the true log-marginal likelihood. Furthermore, similar to our results in Section 5.1, SDE Matching achieves significantly faster convergence than the adjoint sensitivity method and requires approximately $20\times$ less time per training iteration.

### C.2. Robustness of Gradients

We present additional experiments to assess the robustness of SDE Matching.

Using the setup from Appendix C.1, we compare the gradient norms and variances over training time. As shown in Figure 4, SDE Matching consistently exhibits both lower gradient norms and reduced variance compared to the adjoint sensitivity method.

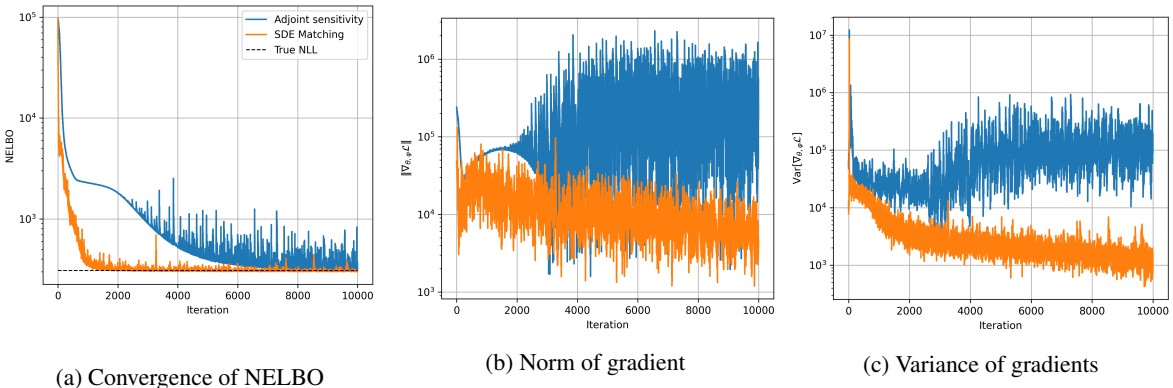

(a) Convergence of NELBO      (b) Norm of gradient      (c) Variance of gradients

*Figure 4.* Linear system training dynamics for Latent SDE trained with adjoint sensitivity method and SDE Matching.

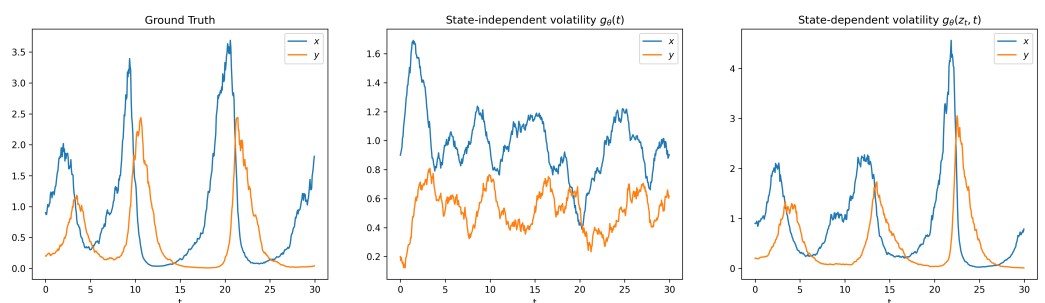

*Figure 5.* Lotka–Volterra system, learned trajectories for SDE Matching with state-independent and state-dependent volatility.

## C.3. Importance of State-Dependent Diffusion Term

We also present an example that highlights the importance of state-dependent diffusion term (volatility), a feature not supported by the previous simulation-free approach ARCTA (Course & Nair, 2023).

To demonstrate this, we design an experiment based on the stochastic Lotka–Volterra system with dynamics:

$$dx = (\alpha x + \beta xy)dt + \sigma x d\mathbf{w} \tag{46}$$

$$dy = (\delta xy + \gamma y)dt + \sigma y d\mathbf{w}. \tag{47}$$

We follow the setup from Course & Nair (2023) (Appendix G.1) with volatility coefficient $\sigma = 0.15$ and use the same parameterization as in Section 5.1.

We then apply SDE Matching to train models with both state-independent and state-dependent volatility functions. As illustrated in Figure 5, the model with state-independent volatility fails to accurately capture the correct trajectory structure, whereas the state-dependent model succeeds.

State-dependant volatility is often a relevant property of many dynamical systems in e.g. finance (Oksendal, 2003), biology (Vadillo, 2019), neuroscience (ElGazzar & van Gerven, 2024), and causal inference (Peters et al., 2022), where Latent SDEs can be applied for modelling. The Lotka–Volterra system which we use in this experiment is an example of such a biological system. In Neural SDEs, state-dependant volatility also leads to dynamics that is more robust to distribution shift (Oh et al., 2024). Finally, the state-dependant volatility is an important contributing factor in our MOCAP experiment (Section 5.2), where the state-dependent SDE Matching and Li et al. (2020) both perform much better than the state-independent ARCTA (Course & Nair, 2023). Therefore, we believe that, together with more general marginals of the posterior process through normalising flows, enabling simulation-free training with state-dependant volatility is also an important contribution of SDE Matching.

# D. Implementation Details

For the experiments presented in Section 5, we follow the setup from Li et al. (2020) for both the 3D stochastic Lorenz attractor and the motion capture dataset. We use the same parameterizations and hyperparameters for the prior process, including the observation models $p_\theta(x_t|z_t)$ and the coordinate-wise diagonal diffusion term $g_\theta(z_t, t)$.

We make a slight modification to the parameterization of the posterior process. First, we do not use an encoder, i.e., a separate network that defines the initial conditions of the posterior process. Second, Li et al. (2020) propose using a time-reversal GRU layer (Cho et al., 2014) to aggregate information from observations into what they refer to as a context. This context is then used to predict the drift term of the posterior SDE via an MLP. To maintain consistency with Cho et al. (2014), we use the same hyperparameters for the GRU and MLP. However, instead of predicting the drift term, we use the MLP to predict the functions $\mu_\varphi$ and $\log \sigma_\varphi$ (Equation (37)) based on the context computed from all observations and the time step $t$. We use diaganal parameterization $\sigma_\varphi$.

For the moving pendulum video experiment, we follow the setup from Course & Nair (2023) for both data generation and the parameterization of the prior process. For the posterior process, we use the same parameterization as in previous experiments, but replace the MLP with a convolutional encoder from Course & Nair (2023) to process image observations.

For optimization, we use the same training hyperparameters, including the Adam optimizer (Kingma, 2014) and the same number of training iterations. However, we do not apply any additional regularization, reweighting, or annealing techniques.

We believe that there exists much more efficient ways to parameterize the posterior process for SDE Matching. However, in this work, our primary goal was to provide a fair comparison with the standard Latent SDE training approach rather than focusing on specific design choices such as prior and posterior model architectures.

