# OpenReview forum: "SDE Matching: Scalable and Simulation-Free Training of Latent Stochastic Differential Equations"
_ICML.cc/2025/Conference — ICML 2025 poster_

### Official Review · Reviewer_JkpK · 2025-02-25

**Overall Recommendation:** 5

**Summary:**

This paper introduces SDE Matching, a novel simulation-free method for training Latent Stochastic Differential Equations (SDEs).  Traditional training of Latent SDEs relies on adjoint sensitivity methods, which are computationally expensive due to numerical integration and backpropagation through SDE solutions. SDE Matching addresses this limitation by leveraging connections with Score and Flow Matching techniques used in generative modeling, by directly parameterizing the marginal posterior distributions of the latent SDE, thereby obviating the need for SDE simulation during training.

## Update after rebuttal
My score remains the same.

**Claims And Evidence:**

The central claims are that SDE Matching enables scalable and computationally efficient training of Latent SDEs, achieving performance on par with adjoint sensitivity methods.  The paper provides experimental evidence on both synthetic (3D stochastic Lorenz attractor) and real-world (motion capture) datasets.  Results demonstrate that SDE Matching achieves comparable or slightly better performance and faster convergence compared to adjoint sensitivity methods.

**Essential References Not Discussed:**

N/A

**Experimental Designs Or Analyses:**

N/A

**Methods And Evaluation Criteria:**

The paper introduces SDE Matching, which involves parameterizing the posterior marginal distribution and deriving a conditional ODE and SDE. The evaluation metrics used include the Negative Evidence Lower Bound (NELBO) for training convergence and Test Mean Squared Error (MSE) on a motion capture dataset for performance evaluation.

**Other Comments Or Suggestions:**

N/A

**Other Strengths And Weaknesses:**

Strengths:
- Innovative and efficient simulation-free training framework for Latent SDEs.
- Significant reduction in computational cost and faster convergence demonstrated empirically.

**Questions For Authors:**

- Can you provide a more detailed comparison of the training and inference wall-time for SDE Matching versus adjoint sensitivity methods, especially for the motion capture dataset?

**Relation To Broader Scientific Literature:**

The proposed methodology builds upon and extends the literature on Latent SDEs, Neural ODEs, and simulation-free training methods such as Score and Flow Matching in generative models.

**Theoretical Claims:**

N/A

---

> ### Author Rebuttal · Authors · 2025-04-01
>
> We are delighted to see that the reviewer finds our approach Innovative, efficient and note the reduction in computational cost of training. Below, we address the questions raised in the review.
>
> **Questions:**
>
> In all our experiments, including the 3D stochastic Lorenz attractor, the motion capture dataset, and additional experiments (see rebuttal to TN6r, Question 3), with the same parameterization and training hyperparameters (such as the number of training steps), SDE Matching requires approximately $5$ times less computation time than the adjoint sensitivity method per iteration. Asymptotically, a single training iteration of SDE Matching takes $\mathcal{O}(1)$ time, while the adjoint sensitivity method takes $\mathcal{O}(L)$, where $L$ is the number of steps in the simulation of the posterior SDE (Eq. 2). However, in practice, the exact training time depends not only on the length of simulation, but also on the evaluation cost of other components, such as the calculation of the prior and reconstruction losses (Eqs. 22 and 24), which affects the actual ratio.
>
> Additionally, as demonstrated in Figure 2, SDE Matching may exhibits significantly faster convergence compared to the adjoint sensitivity method, further accelerating the overall training procedure. When combining the per-iteration improvement and the convergence speed improvement SDE Matching is ~500 times faster than the baseline in the experimental setup from Section 4.1.
>
> At the same time, in SDE Matching, the generative model defined by the prior process (Section 4.1) is exactly the same as in training the Latent SDE using the adjoint sensitivity method. Therefore, the inference time for unconditional sampling is identical for both approaches. However, as discussed in Section 4.4, in the case of forecasting, if we have access to partial observations $x_{t_1}, \dots, x_{t_N}$, SDE Matching enables sampling of the latent state $z_{t_N}$ at the time of the last observation $t_N$ in $\mathcal{O}(1)$ time. In contrast, the conventional parameterization of the posterior process requires simulating the conditional SDE up to  $t_N$. This property of SDE Matching can significantly reduce inference time when observations span a long period. However, in our experiments, we only have access to $3$ samples, all close to $t = 0$. Consequently, most of the inference time is spent simulating the prior process from $t_N$, making the forecasting inference time nearly equal for SDE Matching and the adjoint sensitivity method.
>
> We would like to note that, in principle, SDE Matching allows simulation-free sampling of latent states $z_t$ at arbitrary time steps, potentially enabling fully simulation-free interpolation and forecasting in $\mathcal{O}(1)$ time. However, such inference of the posterior process would require a corresponding training procedure in which the model observes only a few early observations. Otherwise, the posterior process may produce biased samples.

---

### Official Review · Reviewer_JFqV · 2025-03-09

**Overall Recommendation:** 5

**Summary:**

The author(s) proposed SDE-matching, a simulation-free method to fit latent SDE models. The key idea is to use a differentiable normalizing flow method to learn the Markovianization of the posterior SDE and match the probability flow ODE defined by the normalizing flow to get back the SDE.

**Claims And Evidence:**

The claim that the methods can scalably and simulation freely solve latent SDE seems well supported theoretically but not very well tested in experiments.

**Essential References Not Discussed:**

Not off the top of my head.

**Experimental Designs Or Analyses:**

The design is on par with the original latent SDE method and is well-designed. I would appreciate if there are a bit more low dimensional SDE being shown like Fig.1 but it is not necessary.

**Methods And Evaluation Criteria:**

Yes, the method is designed to learn time series with latent SDE, and the paper tested whether the learned latent SDE will predict the time series in both simulated and real data experiments.

**Other Comments Or Suggestions:**

- I think Table 1 is misleading. The proposed method does not provide a gradient over parameters w.r.t. SDE solutions while all other methods do. The proposed method is, in fact, *avoiding* such calculation, but Table 1 could be interpreted as the proposed method could provide a gradient in $O(1)$ time and space which is not true.

- I encourage the author(s) to provide more intuition on why the method could avoid the burden of numerically solving SDE: e.g., is numerically solving SDE provide information that is not necessary, or we implicitly solved the SDE with some NN that are parallelizable so the burden is transformed into training which can be scaled, or any other intuition.

**Other Strengths And Weaknesses:**

- The paper has enough details to reproduce without a code-base.
- Generally well written.
- Clarified why diffusion model can be simulation free as a latent SDE.

**Questions For Authors:**

1) This may not be a real question. But when I tried to reproduce the paper with realNVP, it felt a lot like implementing a physics-informed neural net (PINN), where you have a network approximating solutions of PDE, and in this case, I believe the normalizing flow actually solves the Fokker-Plank equation so maybe it can be seen as a PINN. I wonder if there is a connection.

2) I think in general a difficulty cannot magically go away. I wonder if the author(s) can comment if the difficulty of solving SDE was absorbed into training the normalizing flow. The difficulty here is we need to know the solution of posterior SDE and the author(s) experiment suggests that knowing the marginal seems to be enough, but we still need to approximate it.

3) this is connected to my second question. The authors used only a linear model for posterior SDE. When I tried realNVP it was much harder to train (strongly depends on hyperparameter tuning and takes large epochs, maybe it is just my implementation). To what extent does the success depend on the sampling rate in the data being relatively high so that distributions in between can be approximated with some Gaussians, and if one wants to model low sampling rate data, will the difficulty of learning the (unseen) marginals become more problematic? If time permits, can the author try a different sampling rate in the Lorenz experiment?

4) The objective (23) has $g^{-1}$ in it while $\bar{f}_{\theta,\varphi}$ is itself not scaled by $g$, will this cause numerical issue when $g$ is small?

5) Will eq. 27 imply Gaussian marginals?

6) Will it be more useful to let eq.16 also take the time $(t_1, t_2, ... t_N)$ so some smoothness can be used especially when there are multiple time series taken with different times in data?

7) What if losses in eq. 21 have vastly different scales e.g., the data has very little noise?

**Relation To Broader Scientific Literature:**

- Clarifying what feature made the diffusion model, as a special case of latent SDE simulation free is very insightful.
- Time series modeling with latent SDE is useful in broad scientific fields.

**Theoretical Claims:**

There are not many theories. Majority of the results are known.

---

> ### Author Rebuttal · Authors · 2025-04-01
>
> We are pleased that the reviewer finds our claims to be well supported theoretically, the experiments well designed, and the discussions insightful. We are especially grateful that the reviewer took the time to reproduce our experiments. We sincerely thank the reviewer for such attention to our paper and for the valuable feedback. In response to the comments and questions raised, we provide the following clarifications and commitments.
>
> **Other Comments**
>
> 1. Indeed, in SDE Matching we do not explicitly solve any SDE during training. Instead, we parameterize the posterior process in such a way that allows us to estimate the objective function in $\mathcal{O}(1)$ space and time. The purpose of Table 1 is primarily to reflect the computational budget in terms of number of drift/diffusion term evaluations per training iteration.
>
>     Nevertheless, we would like to note that in SDE Matching, when sampling latent variables $z_t$ to estimate the diffusion loss (Eq. 23), we sample them from the posterior marginals $q_\phi(z_t|X)$, which, by design, are solutions of the posterior SDE (Eq. 19). Thus, during optimization, we are effectively computing gradients with respect to the solutions of the posterior SDE.
>
> 2. In the Latent SDE model, training essentially requires two things: the ability to sample from the posterior marginals $q_\phi(z_t|X)$ and access to the posterior SDE. With the conventional parameterization, we directly define the SDE via its drift, which necessitates numerical simulation in order to sample from the unknown marginals.
>
>     The high-level idea of SDE Matching is to define the posterior process in a different order: we first parameterize a sampler of the marginal distribution and, based on it, derive the posterior SDE. Therefore, we may say that sampling from $q_\phi(z_t|X)$ implicitly solves the posterior SDE.
>
>     It is also important to emphasize that the correspondence between the posterior SDE and its marginals in SDE Matching is exact, not approximate.
>
>
> **Questions:**
>
> 1. There is a connection with PINNs in the tools used to define the model. Like PINNs, we use automatic differentiation to compute time derivatives (Eq. 18) and score functions (Eq. 26). We also employ a partial differential equation (the Fokker–Planck equation) to establish connections between elements of the dynamical system.
>
>     However, unlike typical PINNs that solve PDEs, we use similar mechanics to compute coefficients of an SDE based on its solutions. So, while we believe there are conceptual connections, these are significantly different methods solving different problems.
>
> 2. The difficulty of solving the posterior SDE is reflected in the flexibility of the reparameterization function $F_\phi(\epsilon,t,X)$ (Eq. 16). As discussed in Section 7, the SDE Matching parameterization limits the flexibility of the posterior process.
>
>     If the true posterior process is simple and has tractable marginals, solving it should not be computationally expensive. However, if the posterior is complex, an insufficiently flexible parameterization may not accurately capture the underlying dynamics.
>
>     How complex the posterior latent process needs to be is an open question. Especially in the presence of a flexible observation model.
>
> 3. In this work, we chose a simple parameterization for the posterior process in order to stay close to the setup of Li et al. However, the reviewer’s observations motivate us to explore more expressive parameterizations, and we plan to include those results in a future revision.
>
>     In general, we believe it may be more challenging to marginalize posterior processes of more complex forms, and that the results could be more sensitive to architectural choices in the parameterization network.
>
>     We believe that with denser observations, the posterior process may be better approximated by Gaussian marginals. In contrast, when observations are sparse, intermediate distributions tend to resemble the unconditional marginals, which may have arbitrary shapes.
>
> 4. Small values of  $g_\theta$ may introduce numerical instability, however this issue is not specific to SDE Matching, as the term involving $g_\theta^{-1}$ appears in the ELBO of Latent SDE models in general.
> 5. Yes, Eq. 27 implies that the posterior marginals $q_\phi(z_t|X)$ are conditionally Gaussian.
> 6. Yes, providing time steps to the reparameterization function $F_\phi$ makes perfect sense when working with observations at arbitrary time points.
> 7. In practice, the terms in the objective (Eq. 21) may indeed have different scales (e.g., if the observation model has very low variance). However, the total variational bound remains valid, and all terms should still be minimized to match the true dynamics.
>
> We trust that the clarifications and additional discussions will strengthen your support for acceptance! If accepted, we will update the camera-ready version to reflect this discussion.

---

> > ### Comment · Reviewer_JFqV · 2025-04-01
> >
> > Thank the authors for the clarification!
> > - Right the correspondence between the marginal posterior SDE and its marginals in SDE Matching is exact. I was probably still thinking about the formalism of latent SDE but in SDEmatching the SDE is implicitly defined and marginals matched exactly. Thanks for the clarification.
> >
> > - I think I understand the $O(1)$ space and time being the cost during training. My comment is more because the original papers reported these big-O notations for the task SDE matching tried to not do that for some readers might get confused. I think the table should be there, but I encourage the author to make a note saying this is measured for training time rather than solving SDE and maybe an intuition on how SDEmatching avoided it (again, a difficulty probably won't magically go away and a curious reader would probably want to know how it happened when reading this super encouraging table).
> >
> > - Please include some of the discussions as you see fit in a revision. I am looking forward to it.
> >
> > Again, thank the authors for your time and work. I think it is a very cool paper worth to be seen for many reasons and I have raised my score.

---

> > > ### Author Response · Authors · 2025-04-02
> > >
> > > We will include the clarifications, discussions, and additional experimental results in the revised version of the paper. We sincerely thank the reviewer for the kind words, effort, and constructive feedback. We believe this review was a great help in improving the quality of our work.

---

### Official Review · Reviewer_C8BY · 2025-03-14

**Overall Recommendation:** 4

**Summary:**

This paper improves the method of Course & Nair for variational inference in latent SDEs in 2 ways:  A better recognition network and more flexible marginals through normalizing flows.

**Claims And Evidence:**

The claims are basically that it's a fast and flexible approach.  Also that it's much faster than the Sota in terms of predictive accuracy, Li et al.  However I don't think the speed claim was actually verified empirically (though I believe them).

**Essential References Not Discussed:**

Maybe the Nature version of the Course & Nair work.

**Experimental Designs Or Analyses:**

There are 2 medium-scale experiments.  Not much math.  But it's such a sensible method I don't mind.

**Methods And Evaluation Criteria:**

Just argumentation and a few small-scale experiments.

**Other Comments Or Suggestions:**

"Similarly, interpolation can be performed by inferring only the posterior process dynamics" not clear what this means.

Would love to see more discussion of to what extent matching marginals limits the tightness of the ELBO.

**Other Strengths And Weaknesses:**

Weaknesses:
 - Still using the mocap experiment (introduced in 2019) as the largest experiment.  I guess it's a pain to re-run Li et al but that was 5 years ago, and seems pretty small now.
 - The similarity to Course & Nair.  however, the simplicity and better perf of the new version make this OK.
 - didn't include a time comparison on the mocap experiments.  As it stands, it's not clear what the pareto curve for speed vs accuracy is.

**Questions For Authors:**

Why do you think your method didn't match Li et al's performance on mocap?  Should it be able to, in principle?

**Relation To Broader Scientific Literature:**

This is well-discussed in the paper, pointing out that this is bringing ideas from diffusion models to latent SDE inference (using normalizing flows to match marginals)

**Theoretical Claims:**

The motivations and connections in section 3 are spot on, make sense, and are well-explained.  There aren't really any theorems but I don't think there need to be.

---

> ### Author Rebuttal · Authors · 2025-04-01
>
> We appreciate the reviewer’s positive feedback regarding the speed, simplicity, and sensibility of our method, as well as the discussion of its connections to diffusion models. Below, we address the questions and comments raised in the review.
>
> **References:**
>
> - We will cite the *Nature* paper by Course & Nair in the camera-ready version if accepted.
>
> **Weaknesses:**
>
> 1. To demonstrate the scalability of SDE Matching, we provide additional experiments (please refer to rebuttal to TN6r, Question 3).
> 2. We would like to point out that, compared to the work of Course & Nair, in addition to allowing more flexible marginals of the posterior process, SDE Matching also supports a state-dependent diffusion term $g_\theta(z_t,t)$. This is important in cases where the underlying dynamics exhibit state-dependent volatility.
>
>     To demonstrate this difference, we designed an experiment similar to the setup in Section 4.1, using a stochastic Lotka–Volterra system with a dynamic of the following form:
>
>     $dx=(\alpha x-\beta xy)dt+\sigma xdw$, $dy=(\delta xy-\gamma y)dt+\sigma ydw$
>
>     We then applied SDE Matching to train models with both state-independent and state-dependent volatility functions. Visualisations of the learned trajectories are available at the anonymised link: https://imgur.com/a/jh8sM0Q. It is evident that the model with state-independent volatility fails to capture the correct form of the trajectories.
>
> 3. Please refer to the rebuttal to JkpK for the discussion on training and clarifications on inference (including interpolation).
>
> **Other Comments:**
>
> We would like to highlight that, in contrast to methods such as Flow Matching, both SDE Matching and the adjoint sensitivity method do not match marginals. Instead, they match the distributions of trajectories of the posterior and prior processes. SDE Matching essentially proposes an alternative parameterization of the posterior process that, by design, enables training of the Latent SDE model by minimising the same objective, but in a simulation-free manner.
>
> Nevertheless, the question of the limitations on the tightness of the ELBO in SDE Matching is interesting. Due to the limited space, we may not include rigorous derivations, but we outline the theoretical aspects of this question below.
>
> In the case of the Latent SDE, the variational bound becomes tight when the posterior process matches the true posterior of the prior process.
>
> The prior process (Eq. 15) is given by:
>
> $dz_t=h_\theta(z_t,t) dt+g_\theta(z_t,t)dw$
>
> Given a series of observations $X$, the true posterior process can be derived via Doob’s $h$-transform, yielding:
> $dz_t=\Big[\overbrace{h_\theta(z_t,t)+g_\theta(z_t,t)g^\top_\theta(z_t,t)\nabla_{z_t}\log p_\theta\left(X_t|z_t\right) }^{h_\\theta(z_t,t,X)} \Big] dt+g_\theta(z_t,t)dw,$
>
> where $X_t=\\{x_s:x_s\in X,t\leq s\\}$ denotes the set of future observations from time $t$ onward.
>
> We can also consider the corresponding deterministic process that follows the true posterior marginals $p_\theta(z_t|X)$:
>
> $$\frac{dz_t}{dt}=\bar{h}^\theta(z_t,t,X)=h_\theta(z_t, t)+g_\theta(z_t,t)g^\top_\theta(z_t,t)\left[ \nabla_{z_t}\log p_\theta\left(X_t|z_t\right)-\frac{1}{2}\nabla_{z_t}\log p_\theta\left(z_t|X\right)\right]-\frac{1}{2}\nabla_{z_t}\cdot\left[g_\theta(z_t,t)g^\top_\theta(z_t,t)\right]$$
>
> Now, consider the approximate posterior process (Eq. 19):
>
> $d z_t=f_{\theta,\phi}(z_t,t,X)dt+g_\theta(z_t,t)dw$
>
> The approximate process matches the true posterior process if the drift terms are equal: $f_{\theta,\phi}(z_t,t,X)\equiv h_\theta(z_t,t,X)$, or equivalently, if the corresponding deterministic processes (Eq. 17) match: $\bar{f}^{\theta,\phi}(z_t,t,X)\equiv \bar{h}^\theta(z_t,t,X)$. In terms of the reparameterization function $F_\phi(\epsilon,t,X)$ (Eq. 16) that defines the approximate posterior process, we may say that the variational bound becomes tight, and the approximate posterior process matches the true posterior, if $F_\phi$ learns the solution trajectories of the ODE defined by $\bar{h}_\theta$.
>
> Therefore, with a sufficiently flexible reparameterization function $F_\phi$, the SDE Matching approach should, in principle, be capable of making the variational bound tight.
>
> **Questions:**
>
> SDE Matching shows similar performance to Li et al. as there is an overlap of the confidence intervals. We attribute the slightly better performance of Li et al. to a suboptimal parameterization of the posterior process. In this work, we intentionally kept the parameterization as close as possible to the conventional setup from Li et al. to ensure a fair comparison. In principle, with a sufficiently flexible parameterization, SDE Matching should demonstrate comparable performance.
>
> We will include additional discussions and experimental results with detailed setup explanations in the camera-ready version. We trust these clarifications, highlighting the novelty and scalability of our work, will enhance your support for acceptance!

---

> > ### Comment · Reviewer_C8BY · 2025-04-04
> >
> > I appreciate that a state-dependent noise term is necessary to model state-dependent noise.  This is a valid contribution but seems kind of minor.
> >
> > I also appreciate the stochastic Lotka–Volterra experiments, but would much rather have seen a plot of the ELBO matching a known marginal likelihood!
> >
> > I hadn't appreciated that this paper is optimizing the same ELBO as Li et al.
> >
> > Your argument that you could make the ELBO tight seems reasonable.  But I would have loved to see just a single toy experiment fitting a simple SDE with a known marginal likelihood to show it empirically. (and to see what the gradient variance does during training)
> >
> > Overall I keep my 4 rating because the point of the paper is scalability but the experiments are small.

---

> > > ### Author Response · Authors · 2025-04-07
> > >
> > > We are glad that the reviewer appreciates our theoretical discussion on the tightness of the ELBO in SDE Matching. Below we address additional comments raised and provide experimental validation.
> > >
> > > First, we would like to comment on the importance of state-dependant volatility in Latent SDEs.  State-dependant volatility is often a relevant property of many dynamical systems in e.g. finance [1], biology [2], neuroscience [3], and causal inference [4], where Latent SDEs can be applied for modelling. The Lotka–Volterra system which we use in the rebuttal experiment is an example of such a biological system [2]. In Neural SDEs, state-dependant volatility also leads to dynamics that is more robust to distribution shift [5]. Finally, the state-dependant volatility is an important contributing factor in our MOCAP experiment (Section 5.2), where the state-dependent SDE Matching and [6] both perform much better than the state-independent approach [7]. Therefore, we believe that, together with more general marginals of the posterior process through normalising flows, enabling simulation-free training with state-dependant volatility is also an important contribution of SDE Matching.
> > >
> > > Second, to empirically validate the tightness of the ELBO in SDE Matching we setup a linear stochastic system experiment of the form:
> > >
> > > $dx_t=F(t)x_tdt+L(t)dw,\quad y_t=H(t)x_t+r_t,\quad r\sim\mathcal{N}(0,R(t))$
> > >
> > > For this system, the true log-marginal likelihood is available through the Kalman filter [8]. In our experiment we see that the SDE Matching ELBO indeed converges to the true log-marginal likelihood. Moreover, similar to our experiment from Section 5.1, SDE Matching demonstrates much faster convergence compared to [6] and takes about 20 times less time per training iteration. Additionally, we see that the SDE Matching parameter gradient estimates have consistently lower norm and variance. We provide visualisations of the training dynamic by anonymised link: https://imgur.com/a/D9QXx2a.
> > >
> > > Finally, we would like to demonstrate the difference in performance of SDE Matching compared to adjoint sensitivity approaches in a more high dimensional experiment. We model sequences of 32x32 images, i.e. videos, depicting a moving pendulum. Using identical number of iterations (~20k), SDE Matching successfully learns the dynamics in about 20 minutes on single GPU, while the adjoint sensitivity requires about 30 hours for the same number of iterations and still fails to accurately capture the underlying dynamics. We provide visualisations of generated samples by anonymised link: https://imgur.com/a/8fdV2sD.
> > >
> > > We will add all additional discussions and experiment results, along with detailed descriptions of the setup, in the camera-ready version, if accepted.
> > >
> > > We see no reason why SDE Matching should not scale to even higher dimensional problems, like real world video generation. However, this would require significant computational resources, time and engineering work outside the scope of this paper and rebuttal, so we leave it for future research. Nevertheless, our experiments demonstrate that SDE Matching, compared to the adjoint sensitivity method, has more stable gradients (see experiment with known likelihood above), is robust to scaling of the length of integration (see rebuttal to TN6r, Question 3), requires many times less compute per training iteration and demonstrates faster convergence across a variety of experiments that together results in speedups of several orders of magnitude (see Section 5.1 and the experiment with moving pendulum above). Therefore, we believe, that our experiments clearly demonstrate the scalability of SDE Matching.
> > >
> > > We hope that the additional experiments and clarifications of SDE Matching’s scalability will strengthen your support for our paper.
> > >
> > > [1] Oksendal, Bernt. ”Stochastic differential equations: an introduction with applications.”, Chapter 12
> > >
> > > [2] Vadillo, F. "Comparing stochastic Lotka–Volterra predator-prey models.”
> > >
> > > [3] ElGazzar et al. "Generative Modeling of Neural Dynamics via Latent Stochastic Differential Equations."
> > >
> > > [4] Peters et al. "Causal models for dynamical systems."
> > >
> > > [5] Oh et al. "Stable neural stochastic differential equations in analyzing irregular time series data.”
> > >
> > > [6] Li et al. “Scalable gradients for stochastic differential equations”
> > >
> > > [7] Course et al. "Amortized reparametrization: efficient and scalable variational inference for latent SDEs.”
> > >
> > > [8] Särkkä et al. ”Applied stochastic differential equations”, Chapter 10

---

### Official Review · Reviewer_TN6r · 2025-03-15

**Overall Recommendation:** 2

**Summary:**

This paper builds on the observation that the reverse process in score-based diffusion models can be seen as a neural SDE (as fundamentally it is a process with a parameterized drift). Then the authors try to exploit this connection to develop a simulation-free training scheme for latent SDEs.

**Claims And Evidence:**

The main claim of this paper is that the authors propose a "simulation-free" (definition of which wasn't made clear, see my question below) method for training a latent SDE. The derivation is provided and some experimental results support the claims - although not comprehensive.

**Essential References Not Discussed:**

none

**Experimental Designs Or Analyses:**

Experiments include two main parts, synthetic datasets and the motion capture dataset. While these experiments produce reasonable results, similar papers however include more examples, typically, e.g. moving MNIST or bouncing balls, and similar. I will return to this point in my questions.

**Methods And Evaluation Criteria:**

The paper evaluates the training method on some benchmark problems. Although, it can be definitely said that the evaluation is not comprehensive.

**Other Comments Or Suggestions:**

1) please define simulation-based and simulation-free methods more precisely. Do not assume that every reader knows exactly what you mean by these.

**Other Strengths And Weaknesses:**

See my comments/suggestions - questions.

**Questions For Authors:**

1) The paper essentially tries to formulate the posterior distribution in a latent SDE as a pushforward or some simple distribution. This feels like it is similar to consistency models - can authors elaborate on this comparison?

2) I am not totally convinced that the extra cost of learning the pushforward mapping makes the method really efficient overall w.r.t. simulation-based methods. Figure 2 shows fast convergence, but not in computation time, but in iterations. Could you plot the same graph for computational cost, including any pretraining?

3) Is this method scalable after all? Earlier papers (see my earlier comment) have various image experiments.

4) Is there any scope for theory, assuming that the pushforward measure is accurate enough? Of course, this depends on what's out there for latent SDEs already, but worth discussing.

5) My final comment about this paper is that it falls short of sufficient demonstration of the costs and behaviour of this method. To be clear, I am not interested in to see that this method beats all other baselines - as authors noted, this method can  probably also be used in other baseline methods. However, authors should clearly demonstrate via a series of detailed experiments the behaviour and scalability of this method, even if the results are partially negative. In the current paper, the results are thin and not comprehensive, making it unclear whether this method is sufficiently explored.

**Relation To Broader Scientific Literature:**

This is a continuation of a long line of papers in latent SDEs. Typical methods require long simulations, whereas the current method tries to avoid it; in that sense, it has a distinct purpose.

**Theoretical Claims:**

N/A

---

> ### Author Rebuttal · Authors · 2025-04-01
>
> We thank the reviewers for their valuable feedback, which helps us to improve the paper. Below, we address the questions and comments raised in the review.
>
> **Other Comments:**
>
> We apologise for any confusion regarding terminology. Allow us to provide a clearer definition of the term “simulation-free”.
>
> In the diffusion literature, simulation-free implies that explicit simulation of dynamics (numerical integration) is not required during training. For example, the adjoint sensitivity method [1] requires numerical simulation of the posterior SDE (Eq. 2) to sample latent states $z_t \sim q_\phi(z_t|X)$ to evaluate the loss function (Eq. 5). In contrast, thanks to our alternative parameterization of the posterior process, SDE Matching allows, by design, direct sampling of $z_t$ via inference of the reparameterization function $F_\phi(\epsilon, t, X)$ (Eq. 16), eliminating the need for numerical simulation of the posterior SDE. This makes the training process simulation-free.
>
> **Questions:**
>
> 1. Consistency Models (CM) as originally introduced in [2] learns a function $f_\theta(x_t, t)$ that approximates the solutions of a fixed marginal ODE starting at $x_t$ and ending at some $x_0 \sim p_{data}(x_0)$.
>
>     In SDE Matching, we introduce a reparameterization function $F_\phi(\epsilon, t, X)$ (Eq. 16). Based on $F_\phi$, we derive a conditional ODE (Eq. 17) and then a conditional SDE (Eq. 19) that define the posterior process.
>
>     While both approaches involve a pair of an ODE and its solver, the nature of these ODEs differs: in CM, the goal is to approximate solutions of an unconditional ODE, whereas in SDE Matching we derive an exact ODE as an intermediate step used to define a conditional SDE. Thus, while there is a connection, they are fundamentally different approaches.
>
> 2. SDE Matching does not incur additional learning cost by parameterizing the posterior process via the reparameterization function $F_\phi(\epsilon, t, X)$ (Eq. 16) — this is a central point of our approach.
> Unlike diffusion models, the posterior process in a Latent SDE is not parameter-free. It is a complicated process conventionally defined by the drift function $f_\phi(z_t, t, X)$ in the SDE (Eq. 2), and sampling from the posterior requires simulating this SDE.
>
>     By reparameterizing the posterior process, we neither introduce new parameters nor increase the complexity of the dynamics. Instead, we reparameterise the same dynamics in a different way, enabling direct sampling of latent variables $z_t$ from the posterior marginals without numerically integrating the SDE.
>
>     The key advantage of SDE Matching is that a single training iteration takes asymptotically less time (see rebuttal to JkpK). As discussed in Section 5.1 (lines 320–322), for the 3D Lorenz attractor dataset, one iteration of SDE Matching takes approximately $5$ times less time. Figure 2, while not the main result, additionally shows that SDE Matching not only has faster iterations but also converges more quickly.
>
> 3. To demonstrate the scalability of SDE Matching, we provide two additional experiments.
>
>     First, to evaluate scalability with respect to time, we use the experimental setup described in Section 4.1 and compare the gradient norms of the objective function with respect to the model parameters for both SDE Matching and the adjoint sensitivity method. When integrating over time horizons $T \in {1, 2, 5, 10}$, the adjoint sensitivity method yields the following $\log_{10}$-based gradient norms: $6.26 \pm 0.14$, $6.95 \pm 0.20$, $7.91 \pm 0.23$, and $8.82 \pm 0.28$. This demonstrates that the gradient norms grow exponentially as the time horizon increases.
>
>     In contrast, SDE Matching maintains a stable $\log_{10}$-based gradient norm of $4.92 \pm 0.24$ across all time horizons, indicating better stability for long time series modelling.
>
>     Second, to demonstrate scalability with respect to dimensionality, we designed an experiment where the model learns a sequence of images (a video) depicting a moving pendulum. SDE Matching successfully captures the underlying dynamics and generates realistic samples.
>
>     You can find examples of generated sequence at the anonymised link: https://imgur.com/a/aMfSuHB.
>
> 4. Please refer to rebuttal to Reviwer C8BY (section “Other Comments”), where we discuss the tightness of the variational bound in SDE Matching.
> 5. We hope that the clarifications highlighting the efficiency of our approach, along with additional discussion and experimental results demonstrating the scalability of SDE Matching, will strengthen your support for the acceptance of our paper.
>
> We will include clarifications and additional results, along with detailed explanations of the experimental setups, in the camera-ready version if accepted.
>
> [1] Li et al. “Scalable gradients for stochastic differential equations”
>
> [2] Song et al. “Consistency models”

---

> > ### Comment · Reviewer_TN6r · 2025-04-02
> >
> > Many thanks - can you elaborate how the images are generated? Are these reconstructions or generations where the SDE is simulated and then sampled through the likelihood?

---

> > > ### Author Response · Authors · 2025-04-02
> > >
> > > These images are generated (not reconstructed) samples. In the moving pendulum experiment, to generate the visualizations provided in the rebuttal, we first numerically integrated the learned prior process (the SDE in Eq. 15) and then applied the learned observation model to sample and construct the final outputs (images, in this case).
> > >
> > > SDE Matching successfully learns the dynamics in about 20 minutes on single GPU, while the adjoint sensitivity requires about 30 hours for the same number of iterations and still fails to accurately capture the underlying dynamics. For more details and additional experiments demonstrating the superior stability of SDE Matching's gradients compared to the adjoint sensitivity method, please see the Reply Rebuttal Comment to reviewer C8BY.

---

### Decision · Program_Chairs · 2025-05-01

**Decision:**

Accept (poster)

**Comment:**

This paper builds on the observation that the reverse process in score-based diffusion models can be seen as a neural SDE that can be used for a simulation-free training scheme for latent SDEs. This submission was reviewed by 4 reviewers in the field, and the paper ended up with an average overall recommendation of 4.0 (min: 2, max: 5). The score did not change during the discussion.

The reviewers appreciated the theoretical aspects of the work, while several of the reviewers had concerns related to the experiments. Many of the more minor concerns were resolved during the discussion phase. After the internal discussion, the decision reads that even if the paper would benefit from improvement, it is still over the bar for acceptance.

For the camera-ready, please go through the reviewer comments and try to improve the paper for clarity to ensure proper impact of your work.